# Nup93 regulates breast tumor growth by modulating cell proliferation and actin cytoskeleton remodeling

Simone Bersini[1,2], Nikki K Lytle[3], Roberta Schulte[1], Ling Huang[4], Geoffrey M Wahl[3], Martin W Hetzer[1] ⓘ

**Nucleoporin 93 (Nup93) expression inversely correlates with the survival of triple-negative breast cancer patients. However, our knowledge of Nup93 function in breast cancer besides its role as structural component of the nuclear pore complex is not understood. Combination of functional assays and genetic analyses suggested that chromatin interaction of Nup93 partially modulates the expression of genes associated with actin cytoskeleton remodeling and epithelial to mesenchymal transition, resulting in impaired invasion of triple-negative, claudin-low breast cancer cells. Nup93 depletion induced stress fiber formation associated with reduced cell migration/proliferation and impaired expression of mesenchymal-like genes. Silencing *LIMCH1*, a gene responsible for actin cytoskeleton remodeling and up-regulated upon Nup93 depletion, partially restored the invasive phenotype of cancer cells. Loss of Nup93 led to significant defects in tumor establishment/propagation in vivo, whereas patient samples revealed that high Nup93 and low LIMCH1 expression correlate with late tumor stage. Our approach identified Nup93 as contributor of triple-negative, claudin-low breast cancer cell invasion and paves the way to study the role of nuclear envelope proteins during breast cancer tumorigenesis.**

## Introduction

The nuclear envelope (NE) does not merely represent a protective membrane barrier regulating molecular trafficking between the nucleoplasm and the cytoplasm. Recent findings demonstrate that NE proteins play a crucial role in genome organization and transcription control (Hetzer & Wente, 2009), cell migration (Wolf et al, 2013; McGregor et al, 2016; Graham et al, 2018), and mechanosensing (Uhler & Shivashankar, 2017; Kirby & Lammerding, 2018). Most of these studies have focused on the role of lamins, intermediate filaments generating a dense network underneath the inner membrane of the NE. For instance, mechanical forces applied to the NE

through the linker of cytoskeleton and nucleoskeleton (LINC) were reported to induce lamin A/C accumulation and nuclear stiffening (Guilluy et al, 2014). Mutation or low levels of lamin A/C impaired the formation of the perinuclear actin cap, which protects cells from external mechanical stimuli encountered during confinement and migration (Kim et al, 2017). Moreover, defects in the nuclear lamina were linked to NE rupture in both primary nuclei (Vargas et al, 2012) and micronuclei (Hatch et al, 2013). Despite these recent advances, our knowledge of the functional role of other components of the NE, including the nuclear pore complexes (NPCs), is still limited. NPCs are large protein complexes embedded in the NE composed of 30 different nucleoporins (Nups) whose primary function is the regulation of nucleocytoplasmic transport. Beyond this role, NPCs are now known to directly interact with chromatin and function as regulators of transcription (Ibarra & Hetzer, 2015). In addition, individual nucleoporins contribute to the maintenance of the correct nuclear architecture. For instance, depletion of Nup53, Nup88, and Nup153 was shown to alter the morphology of the nucleus and to impair the assembly of the lamina meshwork (Chow et al, 2012).

A small subset of these nucleoporins (e.g., Nup62, Nup88, Nup98, Nup214, Nup358, and Tpr) has been associated with tumorigenesis, mainly linked to altered nuclear transport activities (Simon & Rout, 2014). Interestingly, the total number of NPCs was reported to be higher in cancer cells when compared with normal tissues, with potential consequences on nucleocytoplasmic transport and gene regulation (Czerniak et al, 1984; Sugie et al, 1994). In this context, a key aspect is that specific nucleoporins have been directly or indirectly associated with cell migration. For instance, Nup62 was shown to localize at the leading edge of migrating cells, actively contributing to cytoskeleton rearrangement and cell motility (Hubert et al, 2009). Nup153 deletion impaired breast cancer cell (BCC) migration through changes in both lamina and cytoskeleton (Zhou & Pante, 2010). In addition, the expression of Nup153 in prostate cancer cells was dependent on estrogen signaling and directly influenced cell motility (Re et al, 2018). Overall, these data suggest that Nups could have unexpected roles in driving cancer cell invasion and influencing their metastatic potential (Friedl et al, 2011). In this context, recent articles reported that specific Nups can

[1]Molecular and Cell Biology Laboratory, The Salk Institute for Biological Studies, La Jolla, CA, USA   [2]Paul F. Glenn Center for Biology of Aging Research at The Salk Institute, La Jolla, CA, USA   [3]Laboratory of Genetics, The Salk Institute for Biological Studies, La Jolla, CA, USA   [4]The Razavi Newman Integrative Genomics and Bioinformatics Core (IGC), The Salk Institute for Biological Studies, La Jolla, CA, USA

Correspondence: hetzer@salk.edu

affect cell cycle progression (i.e., Nup96) (Chakraborty et al, 2008) and others (i.e., Nup93 and Nup153) can drive the expression of cell identity genes through interaction with genome structures called super-enhancers (SE) (Ibarra et al, 2016). These findings are particularly relevant because misregulation of cell cycle and cell identity genes contribute to the cell state reprogramming associated with cancer progression (Wahl & Spike, 2017). In particular, Nup93 might represent an attractive new therapeutic target because Nup93 mutations can increase both cell migration and the expression of epithelial to mesenchymal transition (EMT) markers in breast cancer (Lee et al, 2016). Clinical data also suggest that higher Nup93 expression correlates with reduced breast cancer patient survival (Curtis et al, 2012). To elucidate the mechanisms underlying Nup93 involvement in triple-negative, claudin-low breast cancer progression, we used microfluidic assays, RNAseq, and profiling of Nup93–chromatin interactions. We found that NUP93 silencing impaired BCC 3D migration. This effect correlates with dramatic changes occurring in the actin cytoskeleton (AC), including the formation of compelling stress fibers and large focal adhesions. RNAseq demonstrated that NUP93 silencing up-regulated genes associated with ECM organization and down-regulated genes related to cell migration, EMT, and cell proliferation. More intriguingly, the combination of RNAseq data with profiling of Nup93–chromatin interactions revealed that Nup93 directly interacts with and regulates the expression of a subset of genes involved in AC remodeling, stress fiber formation, and ECM contact. Noteworthy, modulating the expression of one of these genes, LIMCH1, partially reverted the phenotype induced by NUP93 silencing. Our data, confirmed by in vivo experiments and by immunofluorescence of human triple-negative, claudin-low breast cancer samples of different stages provide mechanistic evidence of the role of Nup93 in breast cancer progression and highlight potential novel targets for the development of anti-metastatic therapies.

# Results

### Nup93 modulates cell migration through AC remodeling

Recent evidence suggests that targeting the AC represents a promising strategy in cancer therapy (Foerster et al, 2014). For instance, BCCs respond to cytotoxic natural killer cells by rapidly accumulating F-actin at the immunological synapse and preventing this phenomenon can make cancer cells more susceptible to immune clearance (Al Absi et al, 2018).

Although it is now well accepted that the AC interacts with the nuclear lamina through a wide set of adaptor proteins (Gruenbaum et al, 2005), little is known about a direct connection or regulation of the AC by the NPC. Here, we found that NUP93 silencing in triple-negative, claudin-low MDA-MB-231 (Fig S1A) induced dramatic changes to the AC, including cortical actin thinning (68.0% ± 7.1% versus 100.0% ± 10.3%, data normalized to control cells, P < 0.05) (Fig 1A and B), appearance of paxillin foci (indication of focal adhesion regions) (564.4% ± 48.9% versus 100.0% ± 14.9%, data normalized to control cells, P < 0.001) (Fig 1C and D), and formation of a compelling network of stress fibers (Fig 1E and F). These changes suggest that the cells are experiencing an increased cytoskeletal tension coupled with

focal adhesion remodeling, with potential consequences on their migratory ability. These effects were Nup93 specific because we did not observe the same AC alterations when silencing other nucleoporins (Fig S2). In particular, no major AC changes were quantified after silencing NUP205, whose gene product forms a subcomplex with Nup93 within the NPC core (Ibarra & Hetzer, 2015) (Fig S2). To confirm the specificity of NUP93 silencing, we generated a stable MDA-MB-231 cell line overexpressing an RNAi-resistant NUP93 (i.e., siRes). We found a partial recovery of the total protein level through Western blot (Fig S1B). Similar results were obtained after selection of single colonies of NUP93 silencing–resistant MDA-MB-231. Most importantly, we demonstrated a dramatic decrease in actin stress fibers (labeled with the stress fiber–associated protein LIMCH1) in cells expressing the RNAi-resistant version of NUP93 upon KD (i.e., siRes 93KD) compared with cells in which Nup93 was depleted (Fig 1G and H).

NUP93 silencing also increased the level of FAK (Fig S3A and B) and induced the accumulation of nuclear FAK (Fig S3C and D, P < 0.01). This effect was also observed in NUP155 KD and TPR KD (Fig S2). Although the global increase in FAK might underlie the formation of more stable cell-ECM contacts, the accumulation of nuclear FAK could modify the cell gene expression because FAK can serve as a scaffold protein for several transcription factors (Kleinschmidt & Schlaepfer, 2017). The effect of NUP93 silencing on AC remodeling appears not to be limited to this breast cancer subtype as we observed changes, although more modest, in luminal A MCF-7 (Fig S4), HER2+ MDA-361 (Fig S4), and triple-negative BT-20 (Fig S4) cell lines as well as in non–small cell lung cancer cells H1299 (Fig S5A and B). However, AC modifications seemed to be more substantial in MDA-MB-231 than in MCF-7 (Fig S4) and H1299 (Fig S5A). For this reason, we focused on the triple-negative, claudin-low subtype.

Altering the AC can in principle modify the way cells sense the surrounding microenvironment and migrate through it. To test this hypothesis, we suspended MDA-MB-231 within 3D matrices embedded in multi-channel microfluidic devices (see the Materials and Methods section, Microfluidic invasion assay). Confocal live imaging was used to track the path followed by each individual, viable cell for at least 24 h (Fig 1I). Our results showed a decreased ability of BCCs in which Nup93 has been depleted to invade through a 3D matrix (66.9% ± 6.0% versus 100.0% ± 7.8%, data normalized to control cells, P < 0.01) (Fig 1J and Videos 1, and 2). However, silencing NUP93 did not seem to significantly impact the adhesion ability of the cells, despite the increased expression of focal adhesion proteins (see the Materials and Methods section, Adhesion assay). Indeed, adhesion assay did not show clear differences between NUP93 silenced and control cells (85.4% ± 7.1% versus 100.0% ± 5.9%, data normalized to control cells, P = 0.07) (Fig 1K). Importantly, NUP93 KD in MDA-MB-231 overexpressing the RNAi resistant NUP93 (i.e., siRes 93KD) showed partial functional recovery in the 3D matrix invasion assay compared with wild-type NUP93 silenced cells (87.6% ± 5.2% versus 62.5% ± 2.2%, data normalized to wild-type control cells, P < 0.05) (Fig 1L). No statistical difference was detected comparing siRes NUP93 silenced cells and wild-type control cells (87.6% ± 5.2% versus 100.0% ± 6.8%, data normalized to wild-type control cells, P = 0.305) (Fig 1L). We need to highlight that Nup93 depletion also reduced H1299 cell invasion (Fig S5C, P < 0.01) without

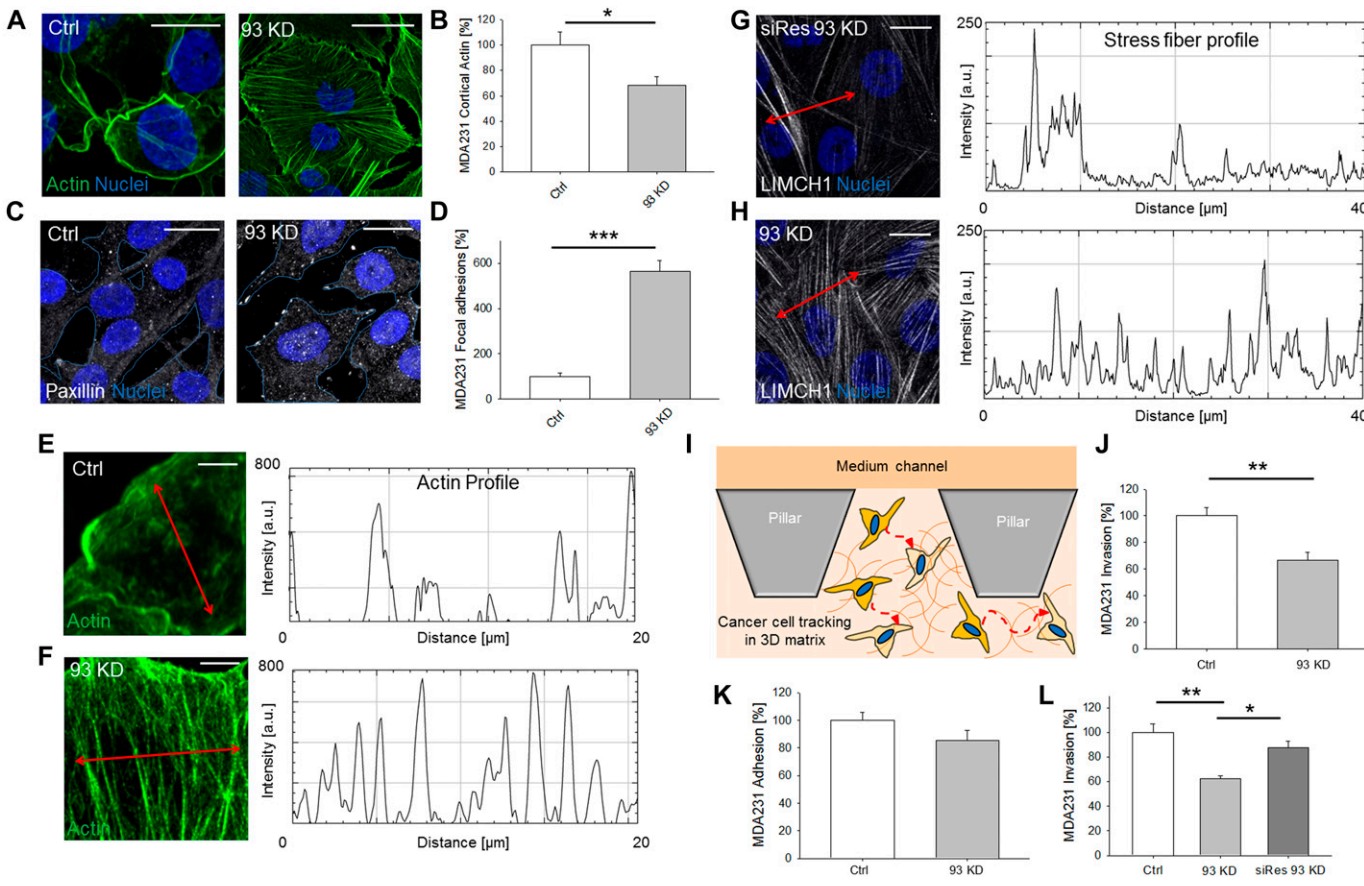

**Figure 1.  Nup93 modulates triple-negative, claudin-low BCC migration through actin cytoskeleton (AC) remodeling.**
**(A, B, C, D, E, F)** *NUP93* KD induces dramatic changes to the cytoskeleton, including thinning of the cortical actin layer (A, B) (*t* test with *P* < 0.05, N ≥ 8 independent regions in three biological replicates, data normalized to ctrl cells), increase in paxillin foci (C, D) (*t* test with *P* < 0.001, N ≥ 4 independent regions in three biological replicates, data normalized to ctrl cells), and formation of a dense network of actin stress fibers (E, F) (details of single cells). Images and intensity profiles are representative of each specific condition. Scale bars: 10 *μm* for (A, C) and 5 *μm* for (E, F). **(G, H)** Partial rescue of Nup93 limits the expression of the stress fiber–related protein LIMCH1. Representative images and intensity profiles. Scale bars: 10 *μm*. **(I, L)** *NUP93* KD impairs breast cancer cell 3D migration. **(I, J, K)** Schematic of the microfluidic assay for 3D single cell tracking (I) and quantification of cell migration (J) (*t* test with *P* < 0.001, N ≥ 100 single cells tracked per condition in three biological replicates, data normalized to ctrl cells) and adhesion (K) (*t* test with *P* = 0.07, N ≥ 25 independent regions in three biological replicates, data normalized to ctrl cells). **(L)** Rescue of Nup93 partially restored cell migration (L) (ANOVA test with *P* < 0.05 [*] and *P* < 0.01 [**], N ≥ 120 single cells tracked per condition in three biological replicates, data normalized to ctrl cells. The *P*-value for ctrl versus siRes 93 KD is *P* = 0.305).

compromising adhesion (Fig S5D, *P* = 0.193). However, because Nup93 depletion induced less clear AC changes in H1299 compared with MDA-MB-231, it is possible that additional mechanisms compromising the migratory ability of these cells are involved upon *NUP93* KD.

Taken together, collected data suggest a potential role of Nup93 in the AC organization and, as a consequence, may impact the cell migratory ability.

## Nup93 depletion does not affect the structural integrity of the NE

The integrity of the NE is controlled by a delicate balance of forces exerted by components of the nucleo- and cytoskeleton. Mislocalization, mutation, or depletion of NE proteins such as Nups can have dramatic consequences on the global structure and functionality of the NE. For example, depletion of Nup153 was reported to alter the distribution of both lamin A/C and Sun1, a protein of the LINC complex connecting cytoskeleton and NE (Zhou & Pante, 2010). Furthermore, depletion of the LINC protein Nesprin-1 induced an

increase in the number of focal adhesion complexes, potentially due to an alteration of the equilibrium of forces between nucleus and cytoskeleton (Chancellor et al, 2010). Based on such evidence, we sought to determine if Nup93 does play a role in the organization of the NE. Surprisingly, although Nup93 is a core component of the NPC, the NPC density on the NE surface was not affected in the absence of Nup93 (94.4% ± 6.7% versus 100.0% ± 2.5%, data normalized to control cells, *P* = 0.485) (Fig 2A and Video 3). We also did not observe modifications in the lamin A coverage (106.2% ± 4.5% versus 100.0% ± 3.8%, data normalized to control cells, *P* = 0.335) (Fig 2B and C), indicating that the macroscopic structure of the NE was not compromised. Similarly, the nuclear distribution of the LINC protein Nesprin-1 was not modified (119.3% ± 32.3% versus 100.0% ± 10.3%, data normalized to control cells, *P* = 0.792) (Fig 2D). However, we found that depletion of Nup93 specifically induced a significant increase in the cytoplasmic content of Nesprin-1 (307.3% ± 76.3% versus 100.0% ± 18.8%, data normalized to control cells, *P* < 0.05) (Fig 2E and F), an event that could be linked to the observed

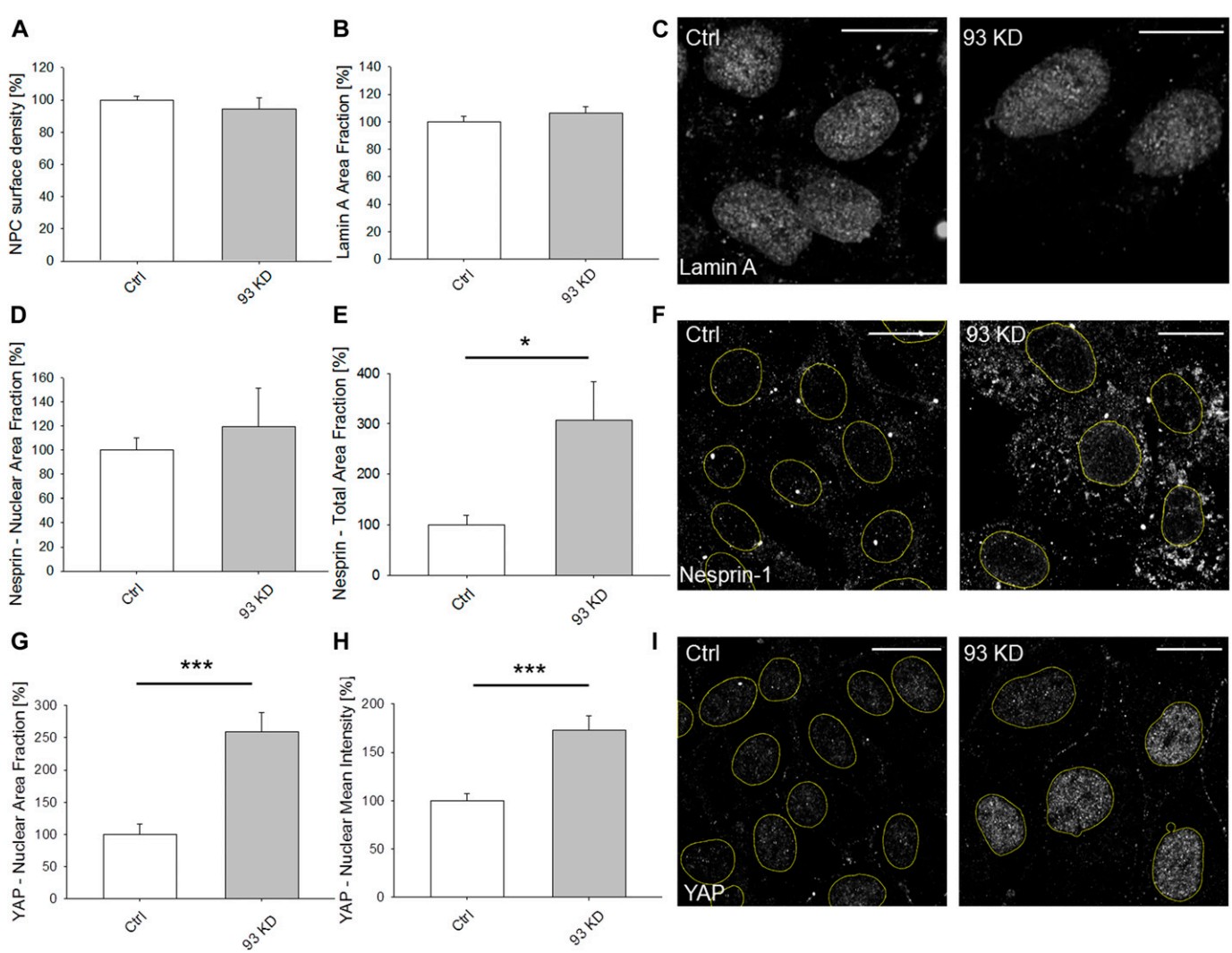

**Figure 2. *NUP93* KD does not affect the structure of the nuclear envelope.**
**(A, B)** Quantification of nuclear pore complex surface density (*t* test with *P* = 0.485, N ≥ 18 independent regions in three biological replicates, data normalized to ctrl cells) and (B) of lamin A area fraction (*t* test with *P* = 0.335, N ≥ 13 independent regions in three biological replicates, data normalized to ctrl cells) after *NUP93* KD.
**(C)** Representative images of lamin A show no structural changes between ctrl and *NUP93* KD BCCs. **(D, F)** Nesprin-1 analysis. **(D, E)** Quantification of nuclear (D) (*t* test with *P* = 0.792, N ≥ 12 independent regions in three biological replicates, data normalized to ctrl cells) and total (E) (*t* test with *P* < 0.05, N ≥ 4 independent regions in three biological replicates, data normalized to ctrl cells) Nesprin-1 area fraction after *NUP93* KD. **(F)** Representative images showing that *NUP93* KD increases Nesprin-1 expression without changes in terms of nuclear localization. Yellow lines are used to indicate cell nuclei. **(G, H, I)** YAP analysis. **(G, H)** Quantification of YAP area fraction (G) and mean fluorescence intensity (H) after *NUP93* KD. *t* test with *P* < 0.001, N ≥ 14 independent regions in three biological replicates, data normalized to ctrl cells.
**(I)** Representative images showing increased nuclear localization of YAP in *NUP93* KD BCCs. Yellow lines are used to indicate cell nuclei. Scale bars: 10 *μ*m.
Source data are available for this figure.

remodeling of the AC and to the presence of stress fibers (Rajgor & Shanahan, 2013). Based on these findings, we hypothesized that cells lacking Nup93 would experience higher forces than control cells because of increased stiffness or intracellular tension. An indirect way to test if the cells are subject to mechanical stimuli is to analyze the subcellular localization of mechanosensitive transcription factors. One of them is Yes-associated protein 1 (YAP). Indeed, it is known that mechanical cues such as substrate rigidity and cell strain can induce YAP translocation to the nucleus (Elosegui-Artola et al, 2017). We found that both nuclear area fraction (258.9% ± 30.1% versus 100.0% ± 16.7%, data normalized to control cells, *P* < 0.001) (Fig 2G) and mean intensity (173.3% ± 14.5% versus 100.0% ± 7.1%, data

normalized to control cells, *P* < 0.001) (Fig 2H) of the YAP fluorescent signal (Fig 2I) were higher in Nup93-depleted cells, suggesting that these cells were indeed subject to higher mechanical stress, as confirmed by the presence of stress fibers and AC remodeling.

Together, our results suggest that the phenotype observed upon Nup93 depletion is not due to either NPC collapse or macroscopic alterations of the NE. Although we cannot exclude that *NUP93* silencing can partially affect the assembly of FG Nups within the NPC, we did not observe mislocalization of nuclear proteins, including lamin A/C, which might suggest a defect in nucleocytoplasmic transport. To further strengthen this point, we considered the nuclear localization of FAK. If changes to the Nup93 sub-complex (which

contains among the others Nup93, Nup205, and Nup188) were responsible for an altered nucleocytoplasmic transport, we would expect a consistent nuclear accumulation of FAK upon depletion of each specific nucleoporin. Conversely, whereas *NUP93* KD induced FAK nuclear translocation, the level of nuclear FAK detected in *NUP205* KD cells was comparable with control cells (Fig S6).

Based on these findings, we then hypothesized that Nup93 could rather be involved in the regulation of genes associated with AC organization and migration.

### Nup93 is required for proper gene expression

Nup93 was recently shown to regulate gene expression in colorectal cancer cells (Labade et al, 2016). We therefore analyzed which transcriptional programs might be misregulated upon Nup93 depletion and how these alterations might explain the altered AC and the impaired cell migration of triple-negative, claudin-low BCCs. RNAseq highlighted the presence of 1,114 differentially expressed (DE) genes when we compared control and Nup93-depleted cells (Fig 3A). Gene ontology analysis revealed that *NUP93* silencing up-regulated biological processes associated with ECM synthesis and organization (e.g., *SERPINH1*, *COL3A1*, *COL5A1*, *FBN1*, *LAMA1*, and *LAMA2*), response to wounding (e.g., *FBLN1*, *MATN2*, *EDN1*, and *DYSF*, indication of intracellular tension) and negative regulation of cell proliferation (e.g., *NRK*, *RARRES3*, and *NR2F2*) (Fig 3B). In addition, Nup93 depletion resulted in the down-regulation of biological processes linked with mesoderm development (e.g., *NF2*, *ITGB4*, and *SNAI1*), positive regulation of cell motility (e.g., *ITGB3*), mesenchyme

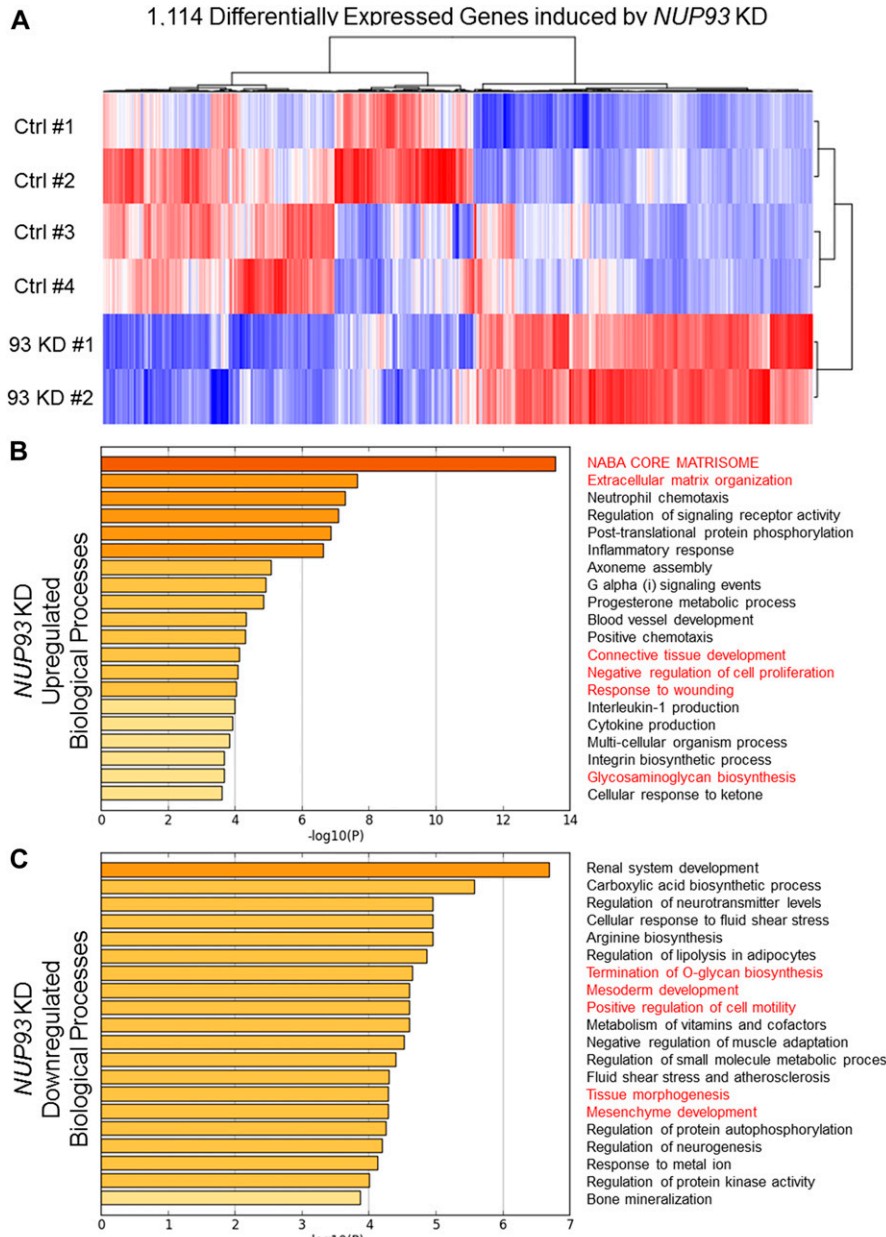

**Figure 3.  *NUP93* KD dysregulates ECM organization, synthesis of glycocalyx proteins, cell motility, and cell proliferation within MDA-MB-231 BCCs.**
**(A)** Heat map of 1,114 significant DE protein-coding genes. RNAseq raw counts were normalized to library depth as cpm. The values were then log-transformed, z-scaled, and hierarchically clustered for heat map visualization. blue = low, red = high (z-score = ±2). **(B, C)** Gene ontology analysis of up-regulated (B) and down-regulated (C) biological processes associated with *NUP93* KD. Analysis performed with Metascape software.

development, neural crest development/differentiation (e.g., *SOX9*, *SEMA4B*, and *FN1*), and EMT (e.g., *SNAI1*, *SOX9*, and *WWTR1*) (Fig 3C). The same analyses were performed on non–small cell lung cancer cells H1299, revealing the presence of a limited subset of DE genes (183) and biological processes in common with triple-negative, claudin-low BCCs. For instance, we observed up-regulation of processes related to ECM organization, collagen metabolism, and connective tissue development/ossification upon *NUP93* silencing (Fig S7). These findings suggest that the effect of Nup93 depletion might not be the same in different cancer cell lines. Similarly, we previously demonstrated that *NUP93* KD induced less pronounced AC modifications in non–small cell lung cancer cells (Fig S5), which showed limited formation of stress fibers and focal adhesions compared with triple-negative, claudin-low BCCs. Finally, we compared the list of identified DE genes with a dataset of molecular predictors of triple-negative breast cancer (GSE71651). This dataset identified 158 DE genes comparing human triple-negative breast cancer and normal breast tissue. We found that nine genes were commonly down-regulated in Nup93-depleted BCCs and human normal tissue. Gene ontology analysis classified some of these genes under cell cycle (e.g., *BUB1*, *CDK1*, and *STMN1*) and wound healing (e.g., *HPSE* and *PKM2*). Importantly, *STMN1* is inversely correlated to stress fiber formation, whereas *PKM2* is a key glycolytic enzyme which can impact cancer cell migration, matrix metalloproteinase expression, and FAK/integrin activation (Yang et al, 2014).

Taken together, the data show that Nup93 either directly or indirectly contributes to the expression of key genes involved in ECM remodeling, cell invasion, and proliferation.

## Nup93 directly regulates genes associated with ECM/cytoskeleton remodeling and cell migration

Our group has recently discovered that Nup93 and Nup153 can influence the cell gene expression by binding genome regulatory structures called SE (Ibarra et al, 2016). Profiling of Nup93–chromatin interactions was then performed to reveal if Nup93 was directly interacting with some of the DE genes identified through RNAseq. To test if Nup93 can interact with genes that are misregulated in its absence, we used a technique called Cut&Run (Skene & Henikoff, 2017). This method allows the identification of interactions between the genomic DNA and a protein of interest similar to ChIPseq with the advantage of a lower background. Cut&Run uses a protein A–fused micrococcal nuclease to release DNA fragments in direct contact with the protein of interest (i.e., Nup93), which has previously been labeled with a primary antibody. We detected 3,272 peaks comparing control and Nup93-depleted cells. Most of these interactions corresponded to introns (~40% peaks), promoters (~27% peaks) and intergenic regions (~26% peaks) (Fig S8). Gene ontology using over-representation analysis enrichment (false discovery rate < 0.05) revealed that these peaks were associated with several biological processes, including "cell junction organization" (which included genes related to cell–cell junctions, focal adhesions, cell migration, and ECM remodeling) and "mitotic cell cycle phase transition" (Fig S8). Because Cut&Run revealed that Nup93 interacts quite frequently with promoter regions, we performed motif enrichment analysis considering all the identified peaks. The top ranked motif (i.e., Jun-AP1) is a transcription factor controlling several biological

processes, including cell proliferation, differentiation, and apoptosis. Although we cannot make any conclusive statements, it would be interesting to test if Nup93 affects downstream transcription through direct interaction or interference with Jun-AP1, as well as with other identified transcription factors (Table S1).

Previous studies demonstrated that Nup93 can interact with SE regions, which are characterized by high density of H3K27ac marks (Ibarra et al, 2016). This interaction is critical to drive the expression of cell identity genes. To test if Nup93 peaks co-localized with H3K27ac peaks, we performed additional Cut&Run analyses finding 168 SE out of which 50 overlapped with the peaks detected upon *NUP93* KD. This overlap was significant using random genomic regions as background (bootstrap $P < 0.001$).

We then combined DE genes found through RNAseq with Nup93 peaks discovered by Cut&Run. Strikingly, we identified 193 genes whose expression was modulated by a direct interaction with Nup93 (Fig 4A). More in detail, 80 genes were up-regulated and 113 genes were down-regulated upon *NUP93* KD. Enrichment analysis showed that up-regulated genes were associated with biological processes including actomyosin structure organization and non-integrin membrane–ECM interactions (Fig S9). These findings are consistent with the AC remodeling and with the modifications in focal adhesions observed in *NUP93* KD cells. On the other side, down-regulated genes were associated with biological processes including actin filament–based process and filopodium assembly (Fig S9). The impaired formation of filopodia and membrane protrusions severely impacts the migration ability of the cells, consistently with the reduced migration observed in *NUP93* KD cells. Analyzing with more detail the list of 193 targets regulated by Nup93, we found genes associated with ECM synthesis, cytoskeleton remodeling, and migration (e.g., *COL11A1*, *LIMCH1*, *EDN1*, *SYNPO2*, and *ITGA6*). Two genes stood out because of their known role in AC organization, *LIMCH1* and *EDN1* (Fig 4B) both coupled with the enhancer marker H3K27ac. Interestingly, we found that one of the identified SE was located in the intronic region of *LIMCH1*. Both these genes could be involved in the observed AC remodeling and stress fiber formation. In particular, LIMCH1 is an actin-related protein whose deletion was associated with decreased stress fibers, reduced focal adhesions, and increased cell migration (Lin et al, 2017). To test the functional significance of LIMCH1 in the context of triple-negative, claudin-low BCCs, we simultaneously depleted Nup93 and LIMCH1 (Fig 4C) with the final goal to verify if this double deletion could partially revert the phenotype of *NUP93*-silenced cells. Microfluidic assay highlighted a partial recovery in the ability of *NUP93+LIMCH1* KD cells to invade a 3D matrix (117.0% ± 7.1% versus 100.0% ± 8.1%, data normalized to *NUP93* KD cells, $P < 0.05$) (Fig 4D). We also observed that *NUP93+LIMCH1* silencing dramatically reduced the number of actin stress fiber peaks (Fig 4E and F).

Overall, our results show that Nup93 directly regulates the expression of a set of genes affecting AC remodeling and cell migration in triple-negative, claudin-low BCCs.

## Nup93 is required for tumor propagation in vivo

Our data on AC remodeling, 3D matrix invasion, and gene expression pointed towards a reduction in the invasive behavior of triple-negative, claudin-low BCCs after *NUP93* silencing. In particular, RNAseq data suggested that cells in which *NUP93* was silenced might

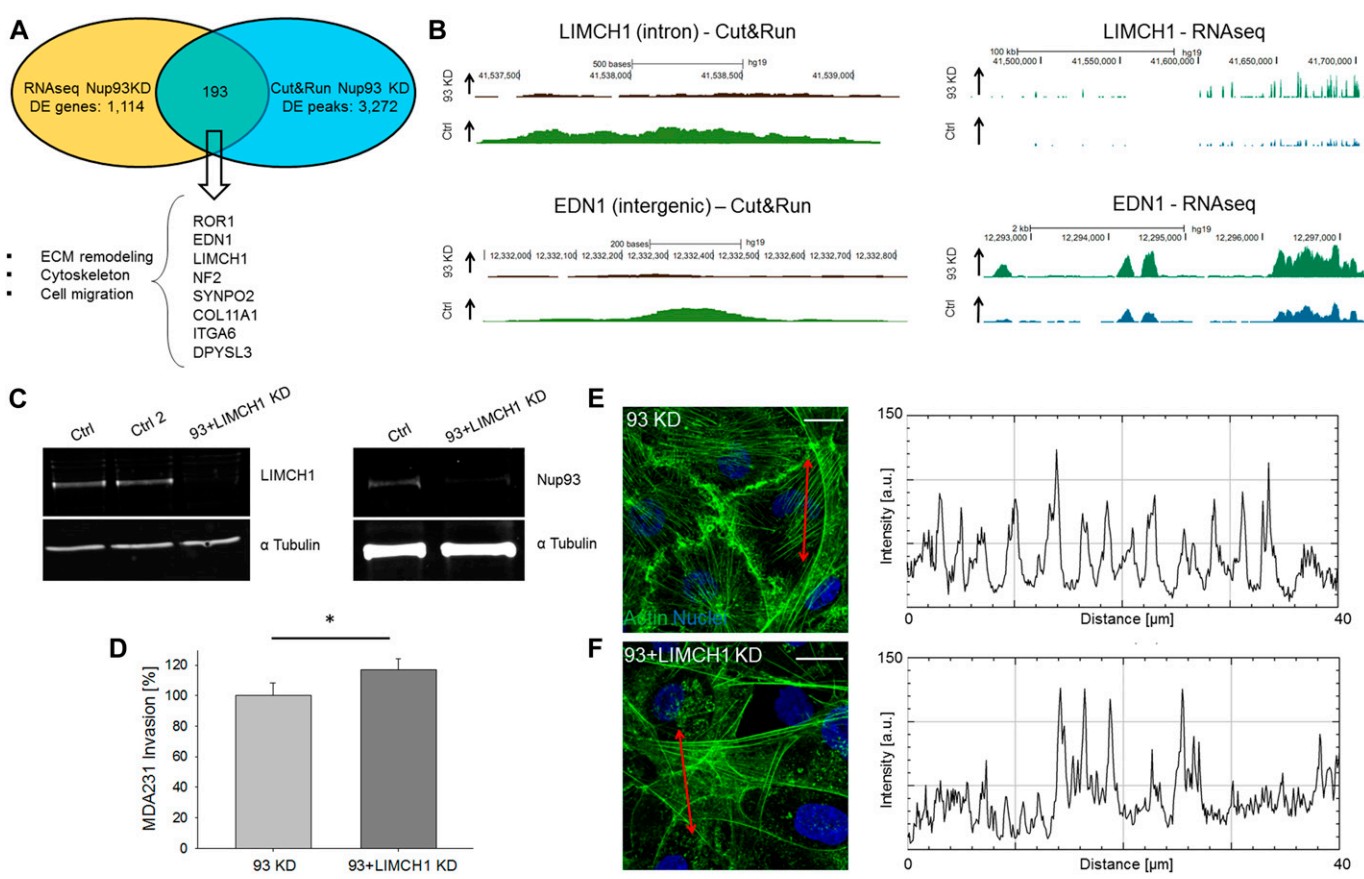

**Figure 4.  Nup93 directly regulates genes associated with ECM/cytoskeleton remodeling, stress fiber formation, and cell migration.**
**(A)** Venn diagram correlating data from RNAseq and Cut&Run (protein-chromatin interactions). *NUP93* KD up-regulates genes associated with stress fiber formation (e.g., *LIMCH1* and *EDN1*). **(B)** Representative images of RNAseq and Cut&Run peaks showing direct regulation of specific genes (representative protein-chromatin interactions at different genomic regions). Tracks were visualized using the University of California Santa Cruz Genome Browser. Tracks of *NUP93* KD and controls are normalized for direct comparison. **(C, D, E, F)** Simultaneous KD of *NUP93* and *LIMCH1* (C) partially restores cell migration compared with *NUP93* KD cells (D) (*t* test with *P* < 0.05, N ≥ 110 single cells tracked per condition in three biological replicates, data normalized to *NUP93* KD cells) and partially impairs the formation of stress fibers (E, F). Representative images and intensity profiles. Scale bars: 10 *μ*m.

be undergoing a mesenchymal to epithelial transition while also experiencing a reduction in cell proliferation. The reduced proliferation was also confirmed in microfluidic invasion assay where *NUP93* silenced BCCs not only migrated significantly less than control cells but also formed smaller micrometastases (i.e., cancer cell aggregates) (Fig S10). To test whether the reduced proliferating ability was indeed due to a defective progression through the cell cycle or to an increase in the apoptosis rate, we generated a stable MDA-MB-231 cell line expressing eGFP upon doxycycline-inducible *NUP93* KD through the miR-E system (Fig 5A). One specific clone (i.e., C5, Fig 5A) was selected for all the in vitro and in vivo experiments detailed below because it showed the highest level of *NUP93* KD. Cells expressing dsRed upon doxycycline-inducible luciferase expression were used as control. *NUP93* KD increased the percentage of cells in G1 (42% ± 0.2% versus 27% ± 0.7%, *P* < 0.001) (Fig 5B) and decreased the percentage of cells in the S phase (51.6% ± 0.3% versus 68.5% ± 0.7%, *P* < 0.001) (Figs 5B and S11A and B). In addition, depletion of Nup93 caused a significant increase in apoptosis (14.2 ± 1.6 versus 5.8 ± 0.7, *P* < 0.001, frequency of annexin V+ cells after FACS analysis) (Figs 5C and S11C and D). Overall, our data pointed towards a potential key role of Nup93 in tumor propagation. To confirm this

hypothesis, we generated mammary tumors within immunocompromised mice (N = 7 per group) by orthotopically transplanting doxycycline-inducible eGFP *NUP93* KD and dsRed control cells at a 1:1 ratio. Doxycycline treatment began when tumors were overtly palpable (10 d after transplantation). After 4 wk, animals were euthanized and primary tumors analyzed. Cells lacking Nup93 were significantly underrepresented in the tumor compared with control cells (4.5% ± 0.7% versus 100% ± 0.8%, *P* < 0.001) (Fig 5D and E [representative FACS plot] and Fig S11E and F). The same ratio was also observed in the few metastatic foci which formed during 4 wk (data not shown). Identical results were obtained when animals were treated with dox immediately after surgery (day 0).

Because *NUP93* silenced cells did not significantly contribute to breast tumor formation and propagation in vivo, we then hypothesized that the level of Nup93 and other proteins directly regulated by Nup93 (e.g., LIMCH1) could be correlated with the progression of the disease in patients. We then collected two samples of stage I and two samples of stage III–IV human triple-negative breast cancer from independent donors. Immunofluorescence analyses revealed that the level of Nup93 (normalized to the level of mAb414, an established marker of nucleoporins) was higher in stage III-IV than in

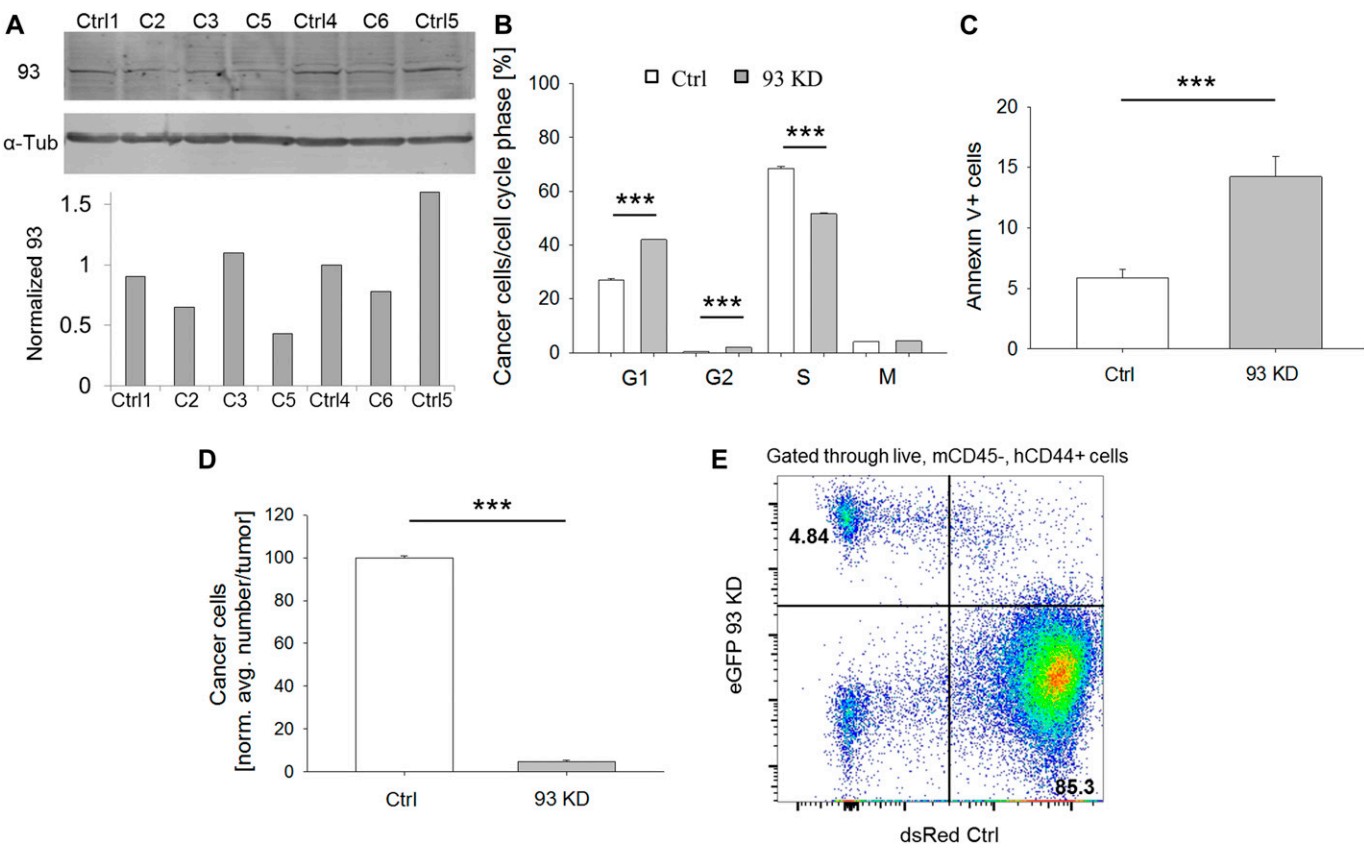

**Figure 5. Nup93 plays a critical role in tumor propagation in vitro and in vivo.**
**(A, B)** *NUP93* KD BCCs (A) proliferate less than control cells (B). C2-C3-C5-C6 represents different *NUP93* KD clones. *t* test with *P* < 0.001 (***), N = 3 independent FACS analyses. **(C)** *NUP93* KD BCCs are more apoptotic than control cells. *t* test with *P* < 0.001 (***), N = 3 independent FACS analyses. **(D, E)** *NUP93* KD reduces the ability of BCCs to form tumors in vivo. Data refer to the "D10 dox" group. Identical results were obtained for the "D0 dox" group. *t* test with *P* < 0.001 (***), N = 7 animals per group, data normalized to ctrl. **(E)** Representative FACS plot of a tumor from the "D10 dox" group showing the ratio between ctrl and 93 KD cells.
Source data are available for this figure.

stage I samples (174.2% ± 16.6% versus 100.0% ± 4.7%, data normalized to stage I, *P* < 0.05) (Fig S12A and C). These qualitative data suggest that the level of Nup93 is increasing irrespectively of the total number of nucleoporins/NPCs. In addition, we stained the same samples for the actin stress fiber–associated protein LIMCH1. Consistent with our RNAseq data showing up-regulation of *LIMCH1* in *NUP93* silenced cells, we found higher levels of this protein in stage I than in stage III-IV samples (Fig S12D).

Together, these results point toward a potential role of the NPC during tumor propagation and disease progression beyond its function of regulator of nucleocytoplasmic transport. It is important to highlight that Nup93 might not be the only nucleoporin involved in breast tumor propagation because preliminary proteomics data showed higher expression of this and other nucleoporins (e.g., Pom121 and Tpr) in MDA-MB-231 than in the non-tumorigenic breast epithelial cell line MCF10A (Fig S13).

## Discussion

We discovered a novel and unexpected role of Nup93 in triple-negative, claudin-low breast cancer. This role goes beyond its function as

structural protein of the NPC and regulator of nucleocytoplasmic transport. Instead, our data reveal that Nup93 is involved in the remodeling of the AC, which has consequences on the ability of the cells to invade through the ECM and contribute to the progression of the tumor. *NUP93* silencing resulted in the formation of a dense network of stress fibers without any indication of NE damage. Most intriguingly, RNAseq suggested that cells in which *NUP93* was silenced acquired both a less proliferative and less invasive behavior, possibly linked to the onset of a mesenchymal to epithelial program. Correlation of RNAseq and Cut&Run data highlighted that Nup93 was directly involved in the transcriptional regulation of a subset of these genes. For instance, Nup93 directly regulated the expression of genes associated with AC remodeling and ECM synthesis (e.g., *LIMCH1* and *EDN1*). As a proof of principle, here we focused on *LIMCH1* as a key regulator of stress fiber formation and AC remodeling. Whereas previous studies already demonstrated the active role of this and other genes in stress fiber formation (Koyama & Baba, 1996; Lin et al, 2017), our data further extend these findings and reveal the surprising discovery of a direct regulation by Nup93. It is important to highlight that we cannot exclude the possibility that another protein localized between Nup93 and the DNA might also be involved in this regulation. We also recognize

that other unexplored targets could reveal additional regulatory functions by Nup93 and further analyses are required to clarify this aspect.

The AC is known to form stable connections with the NE through interactions with the LINC complex (Crisp et al, 2006) or direct indentation of the lamina meshwork (Kim et al, 2017). Then, it is also possible that actin filaments directly bind to specific subunits of the NPC and that modulation of the NPC composition could affect the AC. Although our data point towards chromatin binding of Nup93 as the driving mechanism for the observed phenotype, we did not perform super-resolution microscopy studies to exclude a direct interaction of actin filaments with the NPC and more specifically with Nup93.

A relevant aspect of our study is the possibility to modulate the AC and then compromise the invasion machinery of cancer cells by simply targeting a single nucleoporin. In this context, a recent study demonstrated that estrogen receptor-positive BCCs stimulated with estrogen showed impaired cell invasion and metastases through up-regulation of the actin-related protein EVL. This protein in turn promotes the formation of cortical actin bundles suppressing cell migration (Padilla-Rodriguez et al, 2018). Our findings refer to a different class of breast cancer and highlight a completely different mechanism of action. However, both these studies are in line with the theory that interfering with the cytoskeleton might represent a promising strategy for the development of novel anti-metastatic therapies (Gandalovicova et al, 2017).

Clinical data showed that *NUP93* expression is higher in basal, Her2+, and triple-negative, claudin-low breast cancers than in luminal breast cancer and normal breast tissue (Curtis et al, 2012). These data were qualitatively confirmed by our analyses which showed an increase in Nup93 expression in advanced versus early-stage triple-negative breast cancer patients. Furthermore, clinical data from two independent datasets suggest a potential inverse correlation between the level of *NUP93* and breast cancer patient survival (Metabric dataset [Curtis et al, 2012] [657 patients, log rank *P* = 1.55e-07] which can be found in cBioPortal and MDACC dataset [Usary et al, 2013] [165 patients, log rank *P* = 0.116], which is based on the combination of two studies GEO# GSE25066; Hatzis et al (2011) and GEO# GSE32646; Miyake et al (2012)]). However, it should be carefully considered that only the Metabric dataset reports statistically significant results. Hence, additional basic and clinical data should be collected to explore the possibility that Nup93 or other Nup93-regulated genes could be relevant as therapeutic targets as well as prognostic markers.

Concluding, our study was based on the combination of high-resolution imaging, 3D migration assays, gene expression, chromatin interaction profiling, and in vivo experiments. The former two analyses provided a high-throughput strategy to screen for structural and functional alterations induced by Nup93 dysregulation. At the same time, the combination of two sequencing techniques allowed us to determine the unexpected, direct role of Nup93 in the regulation of key genes associated with AC remodeling, migration and proliferation. Finally, in vivo experiments and analysis of human biopsies confirmed the critical role of Nup93 in tumor propagation. The tumor microenvironment is a complex ecosystem where both cancer and stromal cells contribute to the remodeling of the ECM, which then induces major cytoskeletal changes potentially affecting the invasive potential

of cancer cells (Junttila & de Sauvage, 2013). In this scenario, our findings extend previous studies showing chromatin interactions of Nup93 (Brown et al, 2008; Ibarra et al, 2016; Labade et al, 2016) and reveal a critical role of this protein in BCC migration and breast tumor growth, hence paving the way for the study of other nucleoporins and NE proteins during tumor initiation, EMT, and metastases.

# Materials and Methods

### Cell culture

Triple-negative, claudin-low BCCs MDA-MB-231, normal breast epithelial cells MCF-10A, and non–small cell lung cancer cells H1299 were purchased from ATCC. MDA-MB-231 and H1299 were cultured in DMEM supplemented with 10% FBS and 1% penicillin–streptomycin (herein defined as cancer medium). MCF-10A were cultured in Mammary Epithelial Cell Growth Medium BulletKit (Lonza).

### Silencing

Silencing was performed using siLentFect lipid reagent (Bio-Rad) and validated siRNAs. More in detail, the following sequences were used to design *NUP93* siRNA (used for *NUP93* KD) and luciferase siRNA (used for control cells) (Ibarra et al, 2016):

(*NUP93*) 5′-GCGCUAAUUUACUACUGCA-3′
(Luciferase) 5′-UAUGCAGUUGCUCUCCAGC-3′

Oligos silencing *NUP107*, *NUP133*, *NUP155*, *TPR*, and *NUP205* were commercially obtained from Thermo Fisher Scientific. Oligos silencing *LIMCH1* and *EDN1* (SMART Pool containing four different oligos per target) were commercially obtained from Dharmacon.

Before silencing, the cells were washed with PBS and incubated with Opti-MEM (Thermo Fisher Scientific). Cells were used for any further characterization/assay only after 72 h silencing.

For Nup93 rescue, an MDA-MB-231 cell line overexpressing an RNAi-resistant version of *NUP93* was generated. The *NUP93* sequence recognized by the siRNA was modified by introducing the following mutations: GC<u>C</u>CT<u>T</u>AT<u>A</u>TA<u>G</u>TA<u>G</u>TG<u>G</u>A.

### Generation of stable cell lines

The shERWOOD-predicted RNAi sensor for *hNUP93* (Knott et al, 2014) was inserted into the miR-E plasmid LT3GEPIR (pRRL), whereas the sensor for luciferase was inserted in LT3REPIR in which dsRed had been exchanged for GFP (LT3REPIR) (Fellmann et al, 2013).

*NUP93* miR-E sequence:

TGCTGTTGACAGTGAGCGCCAGGAAAGTGTGGAAGAGAGATAGTGAAGCCA-CAGATGTATCTCTCTTCCACACTTTCCTGATGCCTACTGCCTCGGA

Luciferase miR-E sequence:

TGCTGTTGACAGTGAGCGCAGGAATTATAATGCTTATCTATAGTGAAGCCACAG-ATGTATAGATAAGCATTATAATTCCTATGCCTACTGCCTCGGA

### Microfluidic invasion assay

3D models were developed based on our previous studies (Jeon et al, 2015). Briefly, we designed a microfluidic device with three channels separated by pillars. Each channel was 200 $\mu$m thick. The Smooth-Cast 310 resin was used to create molds. Microfluidic devices were fabricated with poly-dimethyl-siloxane (PDMS) and sterilized in autoclave. Devices were bonded to #1 thickness sterile coverslips (Zeiss) with a plasma cleaner. 1 Mcells/ml BCCs (stained with Vybrant Cell Labeling [Thermo Fisher Scientific] according to the manufacture protocol) were suspended in cancer medium + thrombin (4 U/ml). The cell suspension was then 1:1 mixed with a 5 mg/ml solution of human fibrinogen. 10 $\mu$l of the resulting mix were pipetted into the central channel and incubated at RT for 10 min to polymerize. Next, lateral channels were filled with cancer medium. Cell cultures were kept in humidified incubators (37°C, 5% $CO_2$) for 24 h before starting live imaging on a laser scanning confocal microscope (710; Zeiss) equipped with environmental chamber. The cells were live tracked for 24 h.

### Adhesion assay

Cells were suspended in cancer medium and seeded in eight-well ibidi slides at a density of 3,000 cells/well. The cells were kept in humidified incubators (37°C, 5% $CO_2$) for 2 h and then washed twice with PBS. Next, the cells were fixed and imaged with a laser scanning confocal microscope (710; Zeiss).

### Mass spec: sample preparation and analysis

Cells were washed with cold PBS on ice, scraped off from six-well plates and centrifuged at 450$g$ at 4°C for 3 min. The cells were then resuspended in 100 $\mu$l ice-cold buffer (10 mM Tris–Cl, pH 7.5, 150 mM NaCl, 3 mM MgCl2, and protease inhibitor) by vortexing. Next, 1.5 $\mu$l 10% NP-40 was added to the resuspension and incubated for 5 min on ice. The suspension was then centrifuged at 4,000$g$ at 4°C for 5 min. The resulting supernatant represented the cytoplasmic fraction while nuclei were in the pellet. The pellet was resuspended in 100 $\mu$l buffer (10 mM Tris–Cl, pH 7.5, 150 mM NaCl, 3 mM MgCl2, and protease inhibitor) by tapping the tube. Nuclei were then centrifuged at 4,000$g$ at 4°C for 5 min. Finally, the pellet was dissolved in lysis buffer for Western blot (to check the quality of nuclei fractionation from the cytoplasm) and mass spectrometry. Regarding mass spectrometry, the samples were labeled with Tandem Mass Tag reagents (Thermo Fisher Scientific) and analyzed with a Fusion Orbitrap Tribrid Mass Spectrometer (Thermo Fisher Scientific). Protein and peptide identification were performed with Integrated Proteomics Pipeline–IP2 (Integrated Proteomics Applications).

### RNAseq: library preparation and analysis

RNA was extracted using TRIzol and chloroform following standard protocols. The RNeasy Mini Kit was used to purify RNA (QIAGEN). The samples were incubated with DNAse I to degrade genomic DNA (30 min). RNA was then resuspended in RNAse-free water and its concentration quantified with NanoDrop. The Illumina TruSeq kit was used to prepare cDNA libraries. Quality check was performed using Agilent Tape Station and the amount of cDNA was quantified through Qubit fluorometer. The HiSeq2500 was used to sequence the samples (single-end reading).

Reads were mapped by STAR (v2.5.1b, ref: 10.1093/bioinformatics/bts635. pmid:23104886) to the hg19 reference genome using default parameters. Reads uniquely aligned to exons of RefSeq genes were then quantified by Homer (v4.9.1, ref: PMID: 20513432; http://homer.ucsd.edu/homer/). Differential expression analysis was performed using R DESEq2 package (v.1.18.1, ref: doi: 10.1186/s13059-014-0550-8). Batch effect was added to the design matrix to account for differences introduced by expressing luciferase vectors in cell lines. Significantly DE genes were identified at cutoff of adjusted $P$-value < 0.05 and absolute log2 fold-change > 1. Heat map was generated using R package gplots (https://cran.r-project.org/web/packages/gplots/index.html). Gene ontology figures were generated with Metascape software (http://metascape.org).

### Cut&Run: library preparation and analysis

Libraries were prepared according to published protocols (Skene & Henikoff, 2017). Briefly, the cells were bound to concanavalin-A–coated magnetic beads and incubated with the desired primary antibody after membrane permeabilization with digitonin. After washing, the cells were incubated with protein A and micrococcal nuclease (pA-MN) on ice, and Ca++ was added to start the reaction. DNA fragments were released by nuclease cleavage and finally extracted from the supernatant. Spike-in control DNA was added to each sample for normalization.

Similar to RNA-Seq analysis, reads were mapped by STAR (v2.5.1b, ref: 10.1093/bioinformatics/bts635. pmid:23104886) to hg19 genome but with splicing mode turned off using parameters: "--alignIntronMax 1 –alignEndsType EndToEnd." Reads were also mapped to yeast sacCer3 genome to quantify the number of spike-in reads. Normalization factors were calculated as the ratio of spike-in reads over human reads. To maximize peak signals, reads from biological replicates were first pooled together by Homer (v4.9.1, ref: PMID: 20513432; http://homer.ucsd.edu/homer/) to identify peaks specific to each condition (*NUP93* KD and control, respectively), which were then merged to get a full list of peaks found in both conditions. Finally, the number of reads in a given peak was quantified for each individual sample. R package DESeq2 (v.1.18.1, ref: doi: 10.1186/s13059-014-0550-8) was used to find peaks using an experimental design that was similar to RNA-Seq analysis (batch effect correction for vectors, adjusted $P$-value < 0.05 and absolute log fold-change > 1). DESeq2 normalization size factors were estimated as relative ratios of spike-in normalization factors multiplying relative ratios of total mapped reads. The rationale is to account for differences caused by global shift of peak density between samples (estimated by spike-in ratios) plus sample-specific immunoprecipitation efficiency (estimated by total mapped reads ratios). Peaks were further annotated to its closest associated promoters.

### Immunofluorescence and image acquisition

Samples were washed with warm PBS, followed by 10 min fixation with 3% PFA. After washing with PBS, the samples were incubated with 0.1% Triton-X 100 for 10 min. The samples were then washed

with PBS and incubated with 5% BSA at RT for 1 h, followed by overnight incubation at 4°C with primary antibody diluted in PBS containing 5% BSA. The samples were washed three times in PBS for 5 min each and then incubated with secondary antibody diluted in PBS at RT for 3 h. Finally, the samples were washed three times in PBS for 5 min each. The following primary antibodies and reagents were used for immunofluorescence: 1:100 Acti-Stain 555 (staining for AC), PHDH1, Cytoskeleton; 1:100 anti-human Paxillin, mouse monoclonal, AHO-0492; Thermo Fisher Scientific; anti-human Nup153, SA1 mouse ascites from Dr B Burke; 1:200 anti-human FAK, mouse monoclonal, AHO-1272; Thermo Fisher Scientific; 1:100 anti-human lamin A, goat polyclonal, sc-6214; Santa Cruz; 1:200 anti-human Nesprin-1, mouse monoclonal, MA5-18077; Thermo Fisher Scientific; 1:100 anti-human YAP, mouse monoclonal, sc-101199; Santa Cruz; and 1:200 anti-human LIMCH1, rabbit polyclonal, PA5-64136; Thermo Fisher Scientific. 1:200 Alexa Fluor secondary antibodies were used. Nuclei were counterstained with 1:500 Hoechst.

Samples were imaged with Airy Scan 880 Zeiss microscope setting the same configuration (e.g., laser power, digital gain) for each condition (e.g., control and *NUP93* silenced cells).

### Image analysis and quantification

NPC surface density was quantified using Imaris software (spot function) considering z-projected stacks of Nup153-stained samples. Regarding cortical actin, Fiji software was used to manually measure the width of the actin layer at the cell periphery after AC staining. For actin stress fiber quantification, Fiji software was used to draw 20-$\mu$m long lines within z-projected stacks of AC-stained samples. Representative images were collected from N ≥ 3 independent measurements for each condition. For focal adhesion quantification, images of paxillin-stained samples were thresholded (binary images were obtained by selecting the commands "image, adjust, threshold" in the Fiji software; the same value was set for all images referred to the same condition) and the number of paxillin foci identified through the "particle analysis" command using Fiji software. Regarding quantification of the area covered by FAK, lamin A, Nesprin-1, and YAP, images were processed using Fiji software following established protocols (Bersini et al, 2018) (region of interest selection within z-projected stacks; thresholding [binary images were obtained by selecting the commands "image, adjust, threshold"; the same value was set for all images referred to the same condition]; computation of area fraction or mean intensity value of the fluorescent signal). For invasion assay quantification, the cells were live-imaged for 24 h after cell membrane staining with a fluorescent dye. Imaris software was used to individually track the 3D path of single cells (spot function 16 $\mu$m size, quality 6, max distance 40 $\mu$m, and max gap size 3). Regarding quantification of adhesion, fluorescently labeled cells were fixed, and the number of fluorescent spots in each region of interest was quantified using Imaris software.

### Western blot

Cells were lysed in 1X lysis buffer (2× stock solution: 100 mM Tris–HCl, pH 6.8 [5 ml], + 4% SDS [2 g] + 20% glycerol [10 ml] for a total of 50 ml in DI water) for 10 min on ice, and then centrifuged at 10,000*g* at 4°C for 5 min and the supernatant stored at –80°C until

assayed. The protein content was quantified using BCA protein assay (Thermo Fisher Scientific), loaded on 10% SDS gels (75 $\mu$g protein per lane), and electroblotted onto nitrocellulose membranes. 12% SDS gels were used for endothelin 1 detection (EDN1). Membranes were blocked with 5% BSA in TBST at RT for 1 h under stirring. Then, they were incubated with primary antibody diluted in 5% BSA in TBST at 4°C overnight under stirring. Membranes were washed in TBST for three times, 5 min each under stirring. Finally, the membranes were incubated with secondary antibody compatible with Odyssey Imaging System at RT for 1 h under stirring. After washing three times with TBST, 5 min each under stirring membranes were imaged using Odyssey Imaging System. The following primary antibodies were used: 1:1,000 antihuman EDN1, rabbit polyclonal, PA3-067; Thermo Fisher Scientific; 1:1,000 antihuman LIMCH1, rabbit polyclonal, PA5-64136; Thermo Fisher Scientific; 1:5,000 antihuman alpha tubulin, mouse monoclonal, T5168; Sigma-Aldrich; and 1:500 antihuman Nup93, rabbit, lab developed.

### Human tissue samples

Human triple-negative breast cancer samples were commercially obtained from Indivumed. All samples were collected by Indivumed under informed consent (https://www.indivumed.com/ethical-principles).

Formalin-fixed paraffin-embedded, 10-$\mu$m-thick tissue slices were obtained from triple-negative breast cancer patients (stage I/Ia and stage IIIa/IV). The samples were deparaffinized followed by antigen retrieval using standard protocols. Briefly, the samples were incubated in sodium citrate buffer, pH 8.5, at 80°C for 30 min, cooled down, and then washed twice with PBS. The samples were incubated with blocking solution (1% BSA, 1% fish skin gelatin, and 0.3% Triton X-100) at RT for 1 h. Then, the samples were incubated with primary antibodies at RT overnight, washed three times in PBS for 5 min each, incubated with secondary antibodies at RT for 1 h, and finally washed three times in PBS for 5 min each. The samples were then mounted with Vectashield (Vector Laboratories) and imaged with a Leica SP8 laser scanning confocal microscope. The following primary antibodies diluted in PBS were used: 1:400 antihuman LIMCH1, rabbit polyclonal, PA5-64136; Thermo Fisher Scientific; 1:100 antihuman Nup93, rabbit, laboratory developed; and 1:100 antihuman mAb414, mouse, laboratory developed. 1:200 Alexa Fluor secondary antibodies were used. Nuclei were counterstained with 1:500 Hoechst.

### In vivo analysis of Nup93 depletion

$2 \times 10^6$ MDA-MB-231 cells were orthotopically transplanted into the right fourth mammary gland of 8-wk-old Fox Chase CB17-SCID female mice (strain 236; Charles River). The cells were prepared as a mixture of $1 \times 10^6$ dox-inducible dsRed control cells and $1 \times 10^6$ dox-inducible eGFP *NUP93* KD cells in 25 $\mu$l of DMEM containing 10% FBS, then mixed 1:1 with GFR Matrigel (354230; Corning). Mice were continuously treated with doxycycline (D9891; Sigma-Aldrich, 2 mg/ml in water containing 50 mg/ml sucrose) to induce expression of siRNA starting at the time of transplantation ("D0 dox") or starting when tumors were palpable (10 d post-transplant, "D10 dox"). 4 wk after dox induction, the tumors were harvested and dissociated as

previously described (Dravis et al, 2018). Cell suspensions were stained with Brilliant Violet 421 anti-mouse CD45 (1:500, 103134; BioLegend), PE/Cy7 anti-human CD44 (1:500, 338816; BioLegend), and DAPI (1:1,000) and analyzed by flow cytometry. Analysis was carried out on an LSRII (Becton-Dickinson), and data were analyzed with FlowJo software.

## In vitro proliferation and apoptosis

Dox-inducible dsRed control cells and dox-inducible eGFP *NUP93* KD cells were mixed at 1:1 ratio, and then plated in six-well plates at a density of 40,000 cells per well. The cells were treated with 1.5 $\mu$g/ml doxycycline either 6 or 24 h after seeding (to be analyzed 72 or 48 h after doxycycline exposure, respectively). For proliferation, the cells were treated with 10 $\mu$M Edu for 5 h, collected, then fixed/permeabilized, stained, and analyzed according to the Click-iT Plus Edu Assay (C10634; Thermo Fisher Scientific). DAPI (1:1,000) was used to assess DNA content. For apoptosis, the cells were collected, washed with cold PBS, resuspended in Annexin V Binding Buffer (422201; BioLegend), and stained with Alexa Fluor 647 Annexin V (640912; BioLegend, 2.5 $\mu$l per 5 × $10^5$ cells). DAPI (1:1,000) was used to mark dead cells. Each assay was performed in triplicate for each time point. Analysis was carried out on an LSRII (Becton-Dickinson), and data were analyzed with FlowJo software.

## Statistics

Statistical tests (nonpaired *t* test or one-way ANOVA) were performed with Prism (GraphPad). Details on statistical tests, exact number of independent measurements, and data normalization are reported in each figure caption. Error bars represent SEM. Details on the statistics applied for RNAseq and Cut&Run data analysis are reported in the "RNAseq: library preparation and analysis" and "Cut&Run: library preparation and analysis'" sections, respectively.

## Data and code availability

Data discussed in this publication were deposited in NCBI's Gene Expression Omnibus and are accessible through GEO Series accession number GSE137691. The release of the data was approved by the Salk Institute Institutional Review Board.

All the raw data contained in this article are available upon reasonable request.

# Supplementary Information

# Acknowledgements

S Bersini greatly acknowledges the Paul F. Glenn Center for Biology of Aging Research at The Salk Institute. This work is supported by the National Institutes of Health Transformative Research Award grant R01 NS096786, the Keck Foundation, and the NOMIS Foundation. This work is also supported by Waitt Advanced Biophotonics Core Facility of Salk Institute with funding from NIH-NCI CCSG: P30 014195 and Waitt Foundation; NGS Core Facility and the Razavi Newman Integrative Genomics and Bioinformatics Core Facility of the Salk Institute with funding from NIH-NCI CCSG: P30 014195 and the Chapman Foundation and the Helmsley Charitable Trust; Flow Cytometry Core Facility of the Salk Institute with funding from NIH-NCI CCSG: P30 014195; Mass Spectrometry Core of the Salk Institute with funding from NIH-NCI CCSG: P30 014195; and the Helmsley Center for Genomic Medicine. Mass spectrometry data were analyzed with the precious help of Prof A Buchwalter (University of California San Francisco). We also acknowledge Indivumed for providing human breast cancer samples and we thank Prof CM Perou (University of North Carolina) for critical discussions on breast cancer clinical data.

## Author Contributions

S Bersini: conceptualization, data curation, formal analysis, investigation, methodology, and writing—original draft, review, and editing.
NK Lytle: in vivo experiments and in vitro experiments of cell cycle progression and apoptosis (including design and data analysis).
R Schulte: generation of cell lines for in vitro and in vivo experiments.
L Huang: bioinformatics analyses.
GM Wahl: in vivo experiments and in vitro experiments of cell cycle progression and apoptosis (including design and data analysis).
MW Hetzer: conceptualization, supervision, funding acquisition, and writing—review and editing.

## Conflict of Interest Statement

The authors declare that they have no conflict of interest.

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
