## [Reviewer comments · Life Science Alliance]

Nup93 regulates triple negative breast tumor growth by modulating cell proliferation and actin cytoskeleton remodeling

Simone Bersini, Nikki K. Lytle, Roberta Schulte, Ling Huang, Geoffrey M. Wahl, and Martin W. Hetzer

DOI: 10.26508/lsa.201900623

Corresponding author(s): Prof. Martin W Hetzer (Salk Institute for Biological Studies)

Review timeline:

Submission Date:	2019-12-04
Editorial Decision:	2019-12-05
Revision Received:	2019-12-12
Editorial Decision:	2019-12-16
Revision Received:	2019-12-17
Accepted:	2019-12-19

Scientific Editor: Andrea Leibfried

Transaction Report:

Please note that the manuscript was previously reviewed at another journal and the reports were taken into account in the decision-making process at Life Science Alliance.

No Peer Review Process File is available with this article, as the authors have chosen not to make the review process public in this case.

December 5, 2019

Re: Life Science Alliance manuscript #LSA-2019-00623-T

Prof. Martin W Hetzer
Salk Institute for Biological Studies
Molecular and Cell Biology Laboratory
10010 N. Torrey Pines Road
La Jolla, CA CA

Dear Dr. Hetzer,

Thank you for transferring your manuscript entitled "Nup93 regulates triple negative breast tumor growth by modulating cell proliferation and actin cytoskeleton remodeling" to Life Science Alliance. The manuscript was assessed by expert reviewers at another journal before, and the editors transferred those reports to us with your permission.

The reviewers, and especially reviewer #2, appreciated the quality of the data provided, but they would have expected further reaching mechanistic insight. While a detailed mechanistic understanding is not needed for publication in Life Science Alliance, reviewer #2 provides constructive input on how to strengthen your conclusions, and we would like to invite you to submit a revised version of your manuscript, taking this input into account. Please get in touch in case you would like to discuss individual revision points further.

Thank you for this interesting contribution to Life Science Alliance. We are looking forward to receiving your revised manuscript.

Sincerely,

Andrea Leibfried, PhD
Executive Editor
Life Science Alliance
Meyerhofstr. 1
69117 Heidelberg, Germany
t +49 6221 8891 502
e a.leibfried@life-science-alliance.org

www.life-science-alliance.org

B. MANUSCRIPT ORGANIZATION AND FORMATTING:

December 16, 2019

RE: Life Science Alliance Manuscript #LSA-2019-00623-TR

Prof. Martin W Hetzer
Salk Institute for Biological Studies
Molecular and Cell Biology Laboratory
10010 N. Torrey Pines Road
La Jolla, CA CA

Dear Dr. Hetzer,

Thank you for submitting your revised manuscript entitled "Nup93 regulates breast tumor growth by modulating proliferation and actin cytoskeleton remodeling". I appreciate your response to the concerns raised and the additional data provided, and would thus be happy to publish your paper in Life Science Alliance pending final revisions necessary to meet our formatting guidelines.

- Please upload all figure files as individual ones, including the supplementary figure files; all figure legends should only appear in the main manuscript file
- Please upload the movies you mention
- It would be good to introduce sub-panels and more descriptors in figures S2 and S4 and S6 for more clarity
- Please either change or mention in the manuscript figure legends that the cell displayed in Fig 1A and S6 is the same
- Please provide the source data for Figure 5A and S2
- Please include the mass spec raw data
- Please describe in the M&M section the procedure to arrive at binary images
- Please provide documentation about informed consent for the Indivumed-derived human tissues / include a statement in your manuscript

A. FINAL FILES:

B. MANUSCRIPT ORGANIZATION AND FORMATTING:

Sincerely,

Andrea Leibfried, PhD
Executive Editor
Life Science Alliance
Meyerohofstr. 1
69117 Heidelberg, Germany
t +49 6221 8891 502
e a.leibfried@life-science-alliance.org
www.life-science-alliance.org

3rd Editorial Decision

19 December 2019

December 19, 2019

RE: Life Science Alliance Manuscript #LSA-2019-00623-TRR

Prof. Martin W Hetzer
Salk Institute for Biological Studies
Molecular and Cell Biology Laboratory
10010 N. Torrey Pines Road
La Jolla, CA CA

Dear Dr. Hetzer,

Thank you for submitting your Research Article entitled "Nup93 regulates breast tumor growth by modulating proliferation and actin cytoskeleton remodeling". It is a pleasure to let you know that your manuscript is now accepted for publication in Life Science Alliance. Congratulations on this interesting work.

DISTRIBUTION OF MATERIALS:

Again, congratulations on a very nice paper. I hope you found the review process to be constructive and are pleased with how the manuscript was handled editorially. We look forward to future exciting submissions from your lab.

Sincerely,
